# PyVaporation: A Python Package for Studying and Modelling Pervaporation Processes

**DOI:** 10.3390/membranes12080784

**Published:** 2022-08-15

**Authors:** Denis Andzheevich Sapegin, Aleksei Viktorovich Chekmachev

**Affiliations:** Quantori, 625 Massachusetts Ave., Cambridge, MA 02139, USA

**Keywords:** permeance equation, process modelling, non-ideal transport

## Abstract

PyVaporation—a freely available Python package with an open-source code for modelling and studying pervaporation processes—is introduced. The theoretical background of the solution, its applicability and limitations are discussed. The usability of the package is evaluated using various examples of working with and modelling experimental data. A general equation for the representation of a component’s permeance as a function of feed composition, temperature and initial feed composition is proposed and implemented in the developed package. The suggested general permeance equation may be used for the description of an extremal character of permeance as a function of process temperature and feed composition, allowing the description of processes with a high degree of non-ideality. The application of the package allowed modelling experimental points of various sets of hydrophilic pervaporation data and data on membrane performance from independent sources with a relative root mean square deviation of not more than 9% for flux and not more than 5% for a separated mixture concentration. The application of the facilitated parameter approach allowed the prediction of the components’ permeance as a function of feed concentration at various initial feed concentrations with a relative root mean square error of 3–26%. The package was proven useful for modelling isothermal and adiabatic time and length-dependent pervaporation processes. The comparison of the models obtained with PyVaporation with models provided in the literature indicated similar accuracy of the obtained results, thereby proving the applicability of the developed package.

## 1. Introduction

Throughout past decades, many membrane-separation processes have proven their efficiency and shown distinctive advantages in application to various industrial and scientific challenges. Such processes are often characterized by distinguishable selectivity, relative ease of scaling, and very low energy consumption in comparison with their principal analogues [1]. It is for these, and not only these reasons, that the development and investigation of membrane materials and processes based on them seem to be more and more attractive nowadays to the scientific community and representatives of industry.

Diffusion membrane processes, which operate at a molecular scale, such as nano- and hyperfiltration [1,2], reverse osmosis [1], gas separation [3,4], pervaporation [5], vapour separation [6] and others, are of particular interest due to their ability to solve extremely difficult separation tasks [7,8,9] while retaining advantages characteristic of membrane-based separation. In such processes, the material of the membrane plays a deterministic role in the overall efficiency and performance [1] due to the mechanisms underlying the realization of selective transport. The selective layer in diffusion membranes is usually non-porous; i.e., it does not have stationary voids (pores) through which the penetrant’s molecules are transported—the transport rather occurs through the quasi-stationary free-volume of the selective layer material [10].

Among all of the membrane materials being studied and developed for diffusion-transport applications, ones based on polymers and their composites are some of the most abundant and are considered among the most promising in many applications [1]. The relative ease of manufacture of membranes themselves and modules based on them, and the ability to fine-tune the selective transport properties of the material by varying the chemical structure and morphology of the material [11,12] provide great potential for the design of high-performance polymeric membranes for a particular number of applications. That is why the transport of small molecules through polymers and membranes based on them has been a topic of great scientific and applied interest. 

Transport phenomena in diffusion membranes are often described with a solution-diffusion model (SDM) [13] and as a first-order assumption that may be represented by three consequent steps—the adsorption of the penetrants at the feed side of the membrane, their diffusion through the selective layer of the membrane, and desorption and removal on the permeate side of the membrane. The diffusion of the penetrant molecule through the bulk of the polymer matrix is believed to be the rate-limiting stage of the process, while the adsorption stage usually plays a determining role in the realization of transport selectivity [11]. Individual-component mass transfer rates, or fluxes (amounts of penetrants transported through the unit area of the membrane per time unit), are considered to be one of the most important numerical characteristics of the membrane material when applied to the separation of a particular gaseous or liquid mixture [10]. In terms of the SDM, the flux of the component through the dense membrane is represented as a product of the component’s permeability coefficient ($\Pi_{i}$—the flux of the penetrant observed in the application of the unit driving force to the membrane of a unit length) and a gradient of the component’s chemical potential realized on the membrane [1]. At the same time, the permeability coefficient, in terms of the approach, is considered a product of the solubility and diffusion coefficients of a particular component [14]. Due to the relatively small thicknesses of the typically considered membranes, the gradient of the chemical potential is usually assumed to be a division of a simple difference between the chemical potential of the component at the feed and permeate sides of the membrane and the thickness of the membrane. When dense isotropic membranes are considered, it is usually useful to know the value of the permeability coefficient so that one could calculate the mass transfer rate of the membrane with a particular thickness at a particular driving force value. On the other hand, when composite or asymmetric membranes are considered [15], the permeability coefficient value is not representative due to the effects of the support, and the significant difference between the thickness of the selective layer and the membrane itself; thus for the description of the transport of a particular membrane, permeance values ($P_{i}=\Pi_{i}/\delta$) are usually used.
(1)$$J_{i}=P_{i}\cdot \frac{d\mu_{i}}{dx}\approx \Pi_{i}\cdot \frac{\Delta \mu_{i}}{\Delta \delta }=P_{i}\cdot \Delta \mu_{i}$$
where *J_i_* is the molar flux, mol·m^−2^·s^−1^; $\Pi_{i}$ is the permeability coefficient, mol·m·m^−2^·s^−1^⋅(J·mol^−1^)^−1^, which, in terms of the ideal SDM case, could be presented as the product of corresponding solubility and diffusion coefficients; δ is the thickness of the membrane, m; and *P_i_* is the permeance, or driving force-normalized flux mol·m^−2^·s^−1^⋅(J·mol^−1^)^−1^.

Considering the nature of a particular process, the main flux Equation (1) may be adapted to describe it, accounting for the main factors influencing the chemical potential values [1]. While the driving force could be relatively easily determined for a considered diffusion process, the permeance values may behave in a complex way, which may not be easily predictable with theoretical approaches [16]. The reasons for this lie in the nature of the diffusion transport and the effect of the mixture on the membrane material itself during separation. Phenomena such as penetrant-induced plasticization [17], the coupled transport effect [18], different examples of transport facilitation [19,20,21], and others have been observed in numerous diffusion membrane processes. The high complexity of theoretical approaches applied to describe the interaction between gaseous and liquid mixtures and a polymer matrix usually does not allow providing a comprehensive approach to the prediction of the mass transfer rate in terms of a particular process; hence, researchers and engineers tend to rely on semi-empirical approaches [16,22,23]. Most problems are associated with the fact that permeance in general may be a complex function of the process conditions, especially the composition of the separated mixture and process temperature. Despite the observed complex behaviour of permeance as a function of feed composition, the dependency of permeance on the temperature at a given composition could usually be well described by an Arrhenius-type equation [1]. However, the values of the apparent activation energy of permeation may also significantly depend on the feed composition [24].

Sometimes, when a comparatively short composition range or a particular membrane–mixture pair is considered, the permeance values may not show a significant dependency on the separated mixture composition; this assumption is the easiest approach to the description of membrane performance. Such processes, where the assumption of components’ permeance independency of the particular mixture composition is relevant, may be referred to as ideal, while those where permeance depends significantly on the feed mixture composition are considered non-ideal. 

Among all of the diffusion membrane processes, pervaporation processes tend to show one of the highest degrees of non-ideality [18]. This is usually due to the high affinity of the membrane material to particular penetrants in a separated mixture, complex and non-ideal behaviour between the penetrants and the specifics of the process, which imply direct contact between the membrane and a separated mixture. Pervaporation is defined as a diffusion membrane separation process accompanied by the phase transition of the separated components from liquid to vapour occurring in direct contact with the membrane [1]. The driving force in terms of pervaporation is defined as the difference between the pressure of the saturated vapour of the component in the feed mixture and its partial pressure in the permeate. The pervaporation process is usually implemented by the vacuum degassing of the permeate space, with simultaneous cooling and condensation of the permeate stream, or by the direct removal of the permeate by gas sweeping to lower partial pressures of the permeating components on the permeate side of the membrane. The difficulties in the description of the pervaporation processes due to the high degree of non-ideality not only limit its commercial application, but also trouble researchers when comparing the performance of various membranes in terms of similar processes [25,26]. 

A number of useful semi-empirical models are known [16,27] for the successful description and modelling of pervaporation processes. Most of them usually require the usage of modern computational tools, have limited applicability and are not freely available to the broad scientific community for implementation and usually need to be developed by the researchers for their particular aims. In the past few years, along with commonly known tools used for numerical modelling, Python [28] has become an increasingly popular instrument within the scientific community [29,30] due to its availability, simplicity and robustness. The possibilities provided by Python, along with the well-established culture of open-source solutions and the availability of approaches developed for the semi-empirical modelling of pervaporation processes, provide a perfect basis for the development of a comprehensive tool for the description and modelling of pervaporation processes.

This work aims to introduce an open-source Python-based solution designed to assist researchers in the field of pervaporation membranes and process development called PyVaporation [31]. The solution was developed based on the evaluation of numerous available semi-empirical models [13,16,22,23,27], which were tested during the development of a number of pervaporation membranes [15,20,32] and laboratory- and pilot-scale pervaporation units [15]. This article presents a detailed description of the theoretical bases of the pervaporation process, which were used for the implementation of basic functions supported in PyVaporation, algorithms implemented for basic computations, assumptions and the applicability of the developed module. Descriptions of various examples of models obtained with PyVaporation and their evaluation against real experimental data are provided to show the applicability and limitations of the framework. 

## 2. Theoretical Background

The most difficult task in the modelling of the pervaporation process is determining the value of a component’s flux under particular process conditions. Most semi-empirical approaches that are successfully applied are based on the SDM and represent the flux as a product of permeance and a driving force. In the case of the pervaporation principal equation of transport in diffusion membranes (1), it may be rearranged per the process specifics: the main fraction of the chemical potential difference across the membrane is represented mainly by the difference in the saturated vapour pressure of the component across the membrane [1] (2). Permeance values are usually reported in SI units (mol·m^−2^·s^−1^·Pa^−1^), gas permeance units (GPU—10^−6^·cm_STP_^−3^·cm^−2^·s·cmHg) or mass-related units (kg·m^−2^·h^−1^·kPa^−1^) [10,14].
(2)$$J_{i}=P_{i}\cdot \left(p_{i\;feed}-p_{i\;permeate}\right)$$
where *J_i_* is the molar or mass flux, kg·m^−2^·h^−1^ or mol·m^−2^·h^−1^; *P_i_* is the permeance, kg·m^−2^·h^−1^·kPa^−1^ or mol·m^−2^·s^−1^·Pa^−1^; and *p_i feed_* and *p_i perm_* are the partial pressure of the component’s saturated vapours in the feed and the permeate, respectively, kPa or Pa.

The predominancy of a target component transport is usually characterized by the selectivity (3) and separation factor (4) values. It is worth noting that the separation factor is defined as identical to the relative volatility parameter used to characterize the evaporation of non-ideal mixtures in the processes of distillation columns design [33], and allows one to directly compare the advantage of pervaporation over ordinary evaporation. While selectivity is usually defined by Equation (3), permeances or permeability coefficients may differ significantly between experiments on the transport of individual components and real mixture separation. Thus, two terms are distinguished—real selectivity (ratio of permeances or permeability coefficients measured in terms of mixture separation) and ideal selectivity (ratio of permeances or permeability coefficients measured in terms of an individual component’s transport) [14].
(3)$$S_{i/j}=\frac{P_{i}}{P_{j}}$$
(4)$$\alpha_{i/j}=\frac{C_{ip}/C_{jp}}{C_{if}/C_{jf}}$$
where *P_i,j_* is the molar permeance of components *i* and *j*; and *C_ip_* (*C_jp_*) and *C_if_* (*C_jf_*) represent the concentration of component *i* in the permeate and feed, respectively. Sometimes, for the evaluation of the overall process efficiency, the pervaporation separation index is considered a useful numerical characteristic [34]:(5)$$PSI=J_{\Sigma }\cdot \left(1-\alpha \right)$$
where *J_Σ_* is the overall pervaporation flux, kg⋅m^−2^⋅h^−1^ or mol⋅m^−2^⋅h^−1^

Considering the aforementioned task of the evaluation of a particular pervaporation process, performance comes down to the determination of all components’ partial fluxes under the process conditions. Equation (2) may help us to divide this task into the determination of a driving force value and corresponding permeance values under the considered conditions.

## 3. Driving Force

The estimation of driving force values usually does not cause a great deal of complication. The main parameters influencing the driving force are the temperature and composition of the feed mixture, the composition of the permeate stream and the pressure at the permeate side of the membrane or temperature of the permeate. Even though most of the mixtures for which separation is of interest in the field of pervaporation show significant deviations from Raoul’s law, several useful approaches exist for the calculation of the values of the saturated vapour pressure of the component in a non-ideal mixture. The NRTL [35], Wilson [36], UNIFAC [37] and UNIQUAC [38] models could be successfully applied for the calculation of the partial pressure of the component’s vapours in a broad composition and temperature range, based on the considered mixture composition at a given temperature and with corresponding semi-empirical parameters. These models allow one to determine the values of the components’ activity coefficients, which can be used to calculate the partial pressure of a component in the saturated vapour of a mixture at a given temperature and composition using Equation (6).
(6)$$p_{i\;sat\;m}=\gamma_{i}\cdot p_{i\;sat\;p}$$
where *p_i sat_* is the pressure of the saturated vapour, usually in kPa, and *m* and *p* stand for the mixture and pure component, respectively; and $\gamma_{i}$ is the dimensionless activity coefficient calculated by the means of afore-described models.

The saturated pressure of the pure component usually may be assessed using semi-empirical relations, such as the Antoine (7) and Frost (8) equations.
(7)$$\ln\left(p_{i\;sat\;p}\right)=a_{A}+\frac{b_{A}}{T+c_{A}}$$
(8)$$\ln\left(p_{i\;sat\;p}\right)=a_{F}+\frac{b_{F}}{T}+\frac{c_{F}}{T^{2}}$$
where *p_i sat_* is the pressure of the saturated vapours over the pure component at a given temperature; *a*, *b* and *c* are empirical constants; and the *A* and *F* indexes stand for Antoine and Frost, respectively.

The determination of a component’s partial pressure in the feed mixture is usually assessed using the aforementioned approach. The partial pressure of the component at the permeate side may be determined as a product of the component’s molar fraction in the permeate and the overall pressure at the permeate side, or by using the temperature and composition of the permeate to calculate the partial pressure of the components in a similar way as that for the feed mixture. Usually, these assumptions of the ideality of permeate vapours are valid and do not drastically affect the accuracy of the model.

## 4. Permeance

The most complicated part of the numerical description of flux values’ behaviour for a given polymeric membrane is the determination of a particular component’s permeance under given process conditions. Among the numerous available data on the selective transport properties of various membranes, the observed type of permeance dependency on the process conditions varies significantly. In some cases where, for example, small concentration ranges are considered or a membrane shows extreme swelling stability, it is possible to assume component permeance as a constant over the considered range of conditions [15,32]. However, this is usually not the case. Generally, the permeances of penetrating components vary significantly with the variation in feed concentration [12,20,22,23,24]. The influence of permeating components on the selective layer material tends to alter its structural, morphological and, as a result, transport characteristics [18]. Due to the complexity of existing theoretical models used for the description of polymer-solvent interactions, most theoretical approaches are impractical for applying to the task of permeance value determination [16]. These approaches are usually extremely specific, have significant limitations, and require an amount of experimental data comparable to that required for semi-empirical modelling. For these reasons, semi-empirical approaches based on the solution–diffusion model have found widespread use among specialists in the field of pervaporation process development. While there are a number of semi-empirical models for the description of permeance dependence on process conditions [16,22], the approach suggested by Vier [39] provides the utmost flexibility. As suggested by Vier, the permeance function of multiple process conditions is represented as a product of the permeance functions of individual conditions (9), and provides a general approach for the description of a particular process on the basis of obtained experimental data. Along with the great flexibility, which often occurs, this model implies a relatively high level of uncertainty introduced by the freedom of choice of each function type. As mentioned earlier, the main parameters that usually have a significant impact on the permeance value are the thermodynamic activity of the components in the feed mixture, represented by their concentrations, and feed temperature (11). Sometimes, the activity of a component in the permeate stream may also alter its permeance value; however, this starts to affect permeance values when the permeate pressures are relatively high, which is usually not the case for typical pervaporation units [39]. The permeate pressure, as a rule of thumb, may usually be considered low when its value is significantly lower than the driving force value. Accounting for this, permeance may be represented as a product of the temperature and feed concentration functions (10).
(9)$$P_{i}\left(x_{1},\;x_{2},\;x_{3}\ldots x_{i}\right)=P_{i}\left(x_{1}\right)\cdot P_{i}\left(x_{2}\right)\cdot P_{i}\left(x_{3}\right)\ldots P_{i}\left(x_{i}\right)$$
(10)$$P_{i}\left(x_{F},\;T\right)=P_{i}\left(x_{F}\right)\cdot P_{i}\left(T\right)$$
where *T* is the temperature, *K*; *x_F_* is the feed concentration, mol or wt. %; and *P_i_* is the permeance of component *i*, kg·m^−2^·h^−1^·kPa^−1^ or mol·m^−2^·s^−1^·Pa^−1^.

There are a number of function types that can describe the dependency of permeance on these parameters. The influence of temperature on the permeance value could usually be represented with high accuracy by using an Arrhenius-type relation [1] (11). The inner exponential factor is usually referred to as the apparent activation energy of permeation or transport. In many cases, its value may be independent of the feed concentration; however, a significant amount of data reported indicate that, in general, the apparent energy of transport activation may significantly depend on the feed concentration [24].
(11)$$P_{i}\left(T\right)=A\cdot exp\left(-\frac{E_{a}}{R\cdot T}\right)$$
where *A* is the pre-exponential term, SI, GPU or kg⋅m^−2^⋅h^−1^⋅kPa^−1^; *E_a_* is the apparent energy of transport activation, J⋅mol^−1^; *R* is the gas constant, J⋅mol^−1^⋅K^−1^, and *T* is the temperature, K.

The dependency of permeance on the feed concentration is usually more challenging to establish. The function proposed by Vier for the description of permeance dependence on the feed composition based on Freundlich’s sorption theory [39], as well as common exponential representations [23], may be found to be inapplicable to cases where permeance shows local maxima and/or minima throughout the considered composition range [20]. This phenomenon may usually be explained by the superposition and/or transition between transport mechanisms occurring in the membrane at various feed concentrations. In these cases, describing the complexity of the dependency of permeance on the feed concentration exponential–polynomial representation, as suggested by Wijmans and Baker [22], is very useful. Even though the authors used this approach for the approximation of flux, they were able to account for the complex behaviour of the parameter with the variation in the feed composition. Considering the aforementioned information, representing permeance with the following general function (12) was proposed to account for the possible extremal character of permeance as a function of the feed concentration, and the influence of the feed concentration on the apparent activation energy of permeation. This approach for the representation of permeance was considered useful and was proven to achieve a high level of accuracy (low relative root-mean-square deviation (RMSD, % = root mean square deviation normalized to an average parameter value) when modelling the performance of various membranes (Figure 1).
(12)$$P_{i}\left(x_{F}\;,\;T\right)=\alpha \cdot exp\left(\sum_{i=1}^{n}\;a_{i}x^{i}\right)\cdot exp\left(-\frac{E_{a}\left(x_{F}\right)}{RT}\right)\;=\alpha \cdot exp\left(\sum_{i=1}^{n}\;a_{i}x^{i}-\frac{\sum_{i=0}^{m}\;b_{i}x^{i}}{T}\right)$$

The fits illustrated in Figure 1 show the ability of Equation (12) to describe not only typical diffusion curves (Pervap 4101, b; Pervap 2510, c), but also those of a more complex nature (a). The superposition of two transport mechanisms throughout the considered composition range in the case of SPI membranes [20] leads to the extremal character of each component’s permeance being a function of the feed composition, which definitely should be considered when modelling a process based on these membranes.

Another problem with the modelling of real pervaporation processes is associated with the fact that permeance values rely drastically on the initial feed concentration of the separated mixture [24]. The membrane’s “swelling history” may affect the selective-transport characteristics due to the complex swelling equilibrium occurring in one or multicomponent–polymer systems. In other words, if the membrane is exposed to the mixture where the thermodynamic activity of a high-affinity component is high, it may be plasticized irreversibly by retaining some quantities of this swollen penetrant, which affects the transport mechanisms and selective-transport properties. This is a usual occurrence for commonly used pervaporation membranes based on poly(vinyl alcohol) [24]. Hence, it is extremely important to account for the membrane’s swelling history. This could be achieved while conducting a set of pervaporation experiments at various concentrations and temperatures of interest without the intermediate drying of the membrane to obtain “wet” diffusion curves, and a similar set of experiments with a thorough intermediate drying of the membrane to obtain “dry” diffusion curves. In this work, the authors will refer to permeance as a function of the feed concentration at a fixed temperature as a diffusion curve (it must be noted that, when applied to gas separation, the term diffusion curve may represent the overall quantity transported over a membrane as a function of the processing time [40] and is used in time-lag methods for the determination of diffusion coefficients in polymeric systems [41]).

To consider the effect of the membrane swelling history, the facilitation rate parameter (FR) [12] may be introduced to account for the change in permeance with variations in membrane swelling. As a first-order assumption, the FR parameter may be represented as a function of the initial feed concentration only. In terms of the transport mechanism, this assumption is interpreted as the constancy of the membrane swelling degree over a considered concentration range, while the swelling degree value is dependent only on the concentration of the initial feed mixture being separated. The permeance value could then be calculated using Equation (13), where FR represents the ratio of the permeance values obtained during the experiments with the intermediate drying of the membrane (*P_dry_*, permeance on a “dry” membrane) and without the thorough intermediate drying of the membrane (*P_swelled_*, permeance on a previously swollen membrane) (15).
(13)$$P_{i}=P_{i}\left(x_{F}\right)\cdot FR\left(x_{F0}\right)$$

This approach has proven useful for the prediction of real selectivity values in terms of highly non-ideal pervaporation processes [12], and, despite its roughness, leads to results with acceptable accuracy (Figure 2). For the successful practical application of this assumption, a researcher needs to only measure two diffusion curves throughout a concentration range of interest: one with a thorough intermediate drying of the membrane between each experimental point (“dry”), and the other without the intermediate drying of the membrane (“real”/“wet”). Then, the facilitation rate for each initial concentration may be calculated using Equation (14), given that the considered “dry” point lies inside the “wet” curve’s concentration range. For the opposite case, the FR should be inverted accordingly. The FR for a particular membrane, in some cases, may be correlated with the relative energy distance between the separated penetrants in the Hansen Space (RED) utilizing an Arrhenius-type relation.
(14)$$FR\left(x_{F}\right)=\frac{P_{dry}\left(x_{F}\right)}{P_{swelled}\left(x_{F}\right)}$$

For the illustration of the accuracy of the proposed strategy, the data for Pervap 4100™ membranes [24] in terms of water/ethanol mixture separation with various initial feed concentrations (6–25 wt% water) at 95 °C were modelled. An experimental “wet” curve obtained in the range from an initial feed concentration of 47.4 wt% to a feed concentration of 2.4 wt% water was approximated using Equation (12) and tuned to represent each real curve with varying initial feed concentrations by the introduction of a corresponding FR parameter. The obtained curves, FR parameter and RMSD % values along with experimental data calculated from [24] are illustrated in Figure 2. It should be noted that there are some cases where swelling may cause irreversible changes to the membrane material, which may not be eliminated after drying; for these reasons, the aforementioned assumptions should be validated prior to usage in a particular case. The relatively low values of RMSD illustrated in Figure 2 for each of the modelled curves (b–d) favour the suggestion of the applicability of the FR parameter approach.

## 5. Heat Balance in the Pervaporation Process

Despite the fact that the heat balance does not directly participate in the general transport equations, it plays an important role in the overall process performance. Given that the temperature influences the pervaporation process drastically, the cooling of the heat mixture or energy required for maintaining its temperature. Along with the heat amount to be withdrawn from permeate vapours for condensation, are important factors to be considered when modelling the pervaporation process. In general, for the determination of key process parameters, such as the feed temperature and energy consumption, four parameters need to be determined: evaporation heat of the feed mixture, heat capacity of the feed and permeate mixtures and condensation heat of the mixture at the permeate side. Assuming that components during the process are evaporated and condensed individually, the overall evaporation/condensation heats may be obtained using Equation (15):(15)$$\Delta H_{mixture}=\sum_{i=1}^{n}\;\nu_{i}\cdot \Delta H_{i}$$
where ∆*H_mixture_* is the evaporation/condensation heat of the mixture, kJ⋅mol^−1^; *ν_i_* is the molar fraction of component *i*; and ∆*H_i_* is the evaporation/condensation heat of pure component *i*, kJ⋅mol^−1^.

The corresponding heats of individual components at a temperature of interest may be obtained using the Clapeyron–Clausius equation from corresponding equations describing a component’s saturated vapour pressure behaviour (Equations (16) and (17)).
(16)$$\Delta H_{i}=b_{Ai}\cdot R\cdot ln\left(10\right)\cdot {\left(\frac{T}{T+c_{Ai}}\right)}^{2}—\mathrm{for}\;\mathrm{Antoine}\;\mathrm{equation}$$
(17)$$\Delta H_{i}=-R\cdot \left(b_{Fi}+\frac{2\cdot c_{Fi}}{T}\right)—\mathrm{for}\;\mathrm{Frost}\;\mathrm{equation}$$
where ∆*H**_i_* is the evaporation/condensation heat of pure component *i*, kJ⋅mol^−1^; *T* is the temperature, *K*; *b_i_* and *c_i_* are constants of the saturated vapour equation of component *i*; and the *A* and *F* indexes correspond to the Antoine and Frost equations.

The heat capacity of a multicomponent mixture as a first-order assumption may be represented similarly to the evaporation heat (18). Unlike the case of the driving force calculation, the non-ideality of the mixtures in terms of heat capacity determination generally does not lead to great inaccuracy in the model; however, this fact must be verified when applying the model to a particular case. The dependency of the heat capacity of an individual liquid component may often, with high accuracy, be represented by a polynomial function (19) [32]. In the assumption that the cooling of the vapours, as well as their condensation, occurs on the condenser, the liquid isobaric heat capacities may be used for the estimation of the heat needed for the cooling of the permeate stream with acceptable accuracy. This assumption is only applicable if the permeate remains liquid at the condenser temperature. A general equation for the determination of the heat required for the mixture’s temperature change may be obtained from the differential form of the heat capacity definition (20).

Using these general and widely accepted engineering principles and assumptions [33], one can successfully apply them for modelling non-isothermal pervaporation processes and for the evaluation of the general energy consumption of a particular considered process.
(18)$$C_{p\;mixture}=\sum_{i=1}^{n}\;\nu_{i}\cdot C_{pi}$$
(19)$$C_{pi}=a_{C}+b_{C}\cdot T+c_{C}\cdot T^{2}+d_{C}\cdot T^{3}$$
(20)$$Q_{i}=\int_{T1}^{T2}\;n_{i}\cdot C_{pi}\left(T\right)\cdot dT\;=n_{i}\cdot \left(a_{C}\cdot \left(T_{2}-T_{1}\right)+b_{C}\cdot \frac{\left(T_{2}^{2}-T_{1}^{2}\right)}{2}+c_{C}\cdot \frac{\left(T_{2}^{3}-T_{1}^{3}\right)}{3}+d_{C}\cdot \frac{\left(T_{2}^{4}-T_{1}^{4}\right)}{4}\right)$$
where *C_p mixture_* is the isobaric heat capacity of the mixture, kJ⋅mol^−1^⋅K^−1^; *ν_i_* is the molar fraction of component *i*; ∆*C_pi_* is the isobaric heat capacity of the pure component *i*, kJ⋅mol^−1^⋅K^−1^; *a*, *b*, *c* and *d* are semi-empirical constants of the heat capacity equation; the *C* index stands for heat capacity; *n_i_* is the quantity of the component, mol; and *Q_i_* is heat of cooling or heating KJ.

## 6. PyVaporation Package

PyVaporation (v 1.1.4) is a freely available Python library with an open-source code [31] included in the python package index (PyPi) [42], which allows a user to model basic pervaporation separation processes. Key areas of modelling include the modelling of diffusion curves and the modelling of isothermal and adiabatic (non-isothermal) processes. The general modelling process requires information on the components, mixture and membrane involved. The package allows either the use of pre-defined data or user-created data for the description of components, mixtures and membranes. The solution was intended to be designed in a way that encourages the storage and exchange of the pervaporation experimental and modelling data in a unified and convenient way.

## 7. Main Classes and Methods

To provide a comprehensive understanding of the package’s applicability, limitations and possibilities, a representation of its architecture is presented in Figure 3 in the form of a high-level UML class diagram. A brief description of the main classes, which represent physical and theoretical objects required for modelling the pervaporation experiment employing the PyVaporation package, is provided to increase the transparency of the theoretical background of implementation and provide clarity, which are necessary for a specialist to use the developed solution.

### 7.1. Component

Real compounds are represented within the package as instances of the *Component* class. Components are defined with a name, molecular weight, a corresponding set of constants for the calculation of their individual saturated vapour pressure with Equation (7) or Equation (8) and a set of constants for the calculation of their molar heat capacity with Equation (19). Thus, the system may calculate their saturated vapour pressures, enthalpy of vaporization (Equations (16) and (17)) and heat capacity at a given process temperature by calling corresponding methods. A user may specify the type of the equation intended for the calculation of saturated vapour pressures and evaporation heats by filling a string-type parameter of the *VapourPressureConstants* object used for the creation of the component. Although the Frost equation may, in some cases, provide a better fitting for the vapour pressure data, due to the data’s abundance, the Antoine type of calculation is set as default.

There are several pre-defined components that are commonly separated through pervaporation, and are stated directly as attributes of the *Components* class. The corresponding parameters of these components were validated against experimental data prior to inclusion in the package. An example of a new component’s creation from pre-defined and user-specified parameters and the calculation of some of its properties is presented below (Figure 1):

### 7.2. Mixture

Mixtures in PyVaparation are binary and defined with a name, a set of NRTL parameters and two associated components. The order of components in the mixture is significant as it affects the calculation of their saturated vapour pressure throughout the mixture [35]. The *Composition* class is used for specifying the composition of the mixture, and a *Composition* object is defined by the weight or molar fraction of the first component of the mixture and its type (weight or molar). Given a mixture, the type of the *Composition* object can be mutually converted, which makes it easier for researchers to work with data of different types. Some of the most abundant mixtures, the separation of which is of interest to researchers in the field of pervaporation membranes, and processes’ development are also pre-defined in PyVaporation, and are accessible as methods of the *Mixtures* class. The calculation of the VLE equilibrium at 333.15 °K for a water/ethanol mixture, along with its definition, is provided in Figure 2. Despite the fact that there are several tools available for similar calculations, to create a robust and easy-to-use solution for a researcher, the NRTL model’s calculation was implemented directly in the package.

### 7.3. Membrane

PyVaporation aims to provide a possibility for a researcher to perform semi-empirical modelling of the process; thus, the membranes, as instances of the *Membrane* class, are defined with experimental data. There are two types of objects used for the representation of experimental data for a considered membrane—*IdealExperiment* and *DiffusionCurve*.

*IdealExperiment* objects are purposed to store information on the transport properties of the membrane in relation to the individual component in terms of the modelled process, assuming its ideality, including the name of the component, temperature of the experiment, permeance value and apparent activation energy of the component’s permeation. While being optional, the apparent activation energy of a component may be calculated if data on the component at two or more different temperatures are provided based on linear least-squares regression. *IdealExperiment* objects are used for modelling ideal processes, where the permeance values for components do not change significantly over the concentration range considered. To provide a comfortable way of working with different permeance units (SI, GPU or kg⋅m^−2^⋅h^−1^⋅kPa^−1^), permeance values are represented as instances of the *Permeance* class defined with a value and units. The class allows conversion between the supported units by calling the *convert()* method, with the corresponding parameters (Figure 3). *IdealExperiment* objects are associated with the object of the *IdealExperiments* class, which could be used to define a membrane and is loaded from a .csv file.

To provide a system with information about the dependence of permeance values on both the feed mixture composition and temperature, a *DiffusionCurve* object for a membrane must be specified. As mentioned earlier, in terms of the present work, a diffusion curve is defined as a function of component permeances on the separated mixture feed composition at a constant temperature. Following this definition, a *DiffusionCurve* object is created based on the information regarding the separated mixture, the name of the membrane that was used in the experiment, the temperature at which the curve was obtained, a list of feed concentrations in the curve and the corresponding components’ permeances and/or partial fluxes. A user may specify only a list of the components’ partial fluxes or permeances, and the corresponding parameters will then be automatically calculated based on the specified curve temperature using the NRTL model and Equation (2) under the assumption of zero permeate pressure. It must be noted that, to obtain accurate modelling results, a user should consider the type (“wet” or “dry”) of the curve provided, as permeance values may depend significantly on membrane swelling induced by contact with a feed mixture with a particular composition. Process parameters such as the permeate temperature and pressure may also be included to increase the accuracy of the permeance calculation from corresponding partial flux values. The parameters stored within *DiffusionCurve* objects may be plotted as a function of the feed composition utilizing the *plot()* method. Some methods for the calculation of derived parameters, such as PSI or the separation factor, are defined for *DiffusionCurve* objects directly. The diffusion curves obtained under similar conditions with variable process temperatures are united inside a *DiffusionCurveSet* object, a list of which may be used to define the membrane. *DiffusionCurveSet* objects, as *IdealExperiments* objects, may be loaded into the system from .csv files.

Overall, a *Membrane* within the PyVaporation package may be defined with either an *IdealExperiments* object, a list of *DiffusionCurveSet* objects or their combination and a name. The package allows loading experimental data provided in the appropriate format to create a *Membrane* object for further use from a directory with a stated structure. This approach provides a researcher with the possibility to store and accumulate experimental data associated with a particular membrane in a single organized file, which could be easily used to model the membrane’s performance.

There are some useful methods defined for the *Membrane* class purposed for the calculation of the apparent activation energy of a particular component, permeance of a particular component at a specified temperature (Equation (11)), based on *IdealExperiments* object, calculation of ideal selectivity values (Equation (3)) of two components at a given temperature and the estimation of a pure component flux of the component at a given temperature (Equation (2)). An example of loading and working with a *Membrane* object is provided below (Figure 4). Pervap 4101, along with Pervap 4100 [24], Pervap 2510 [23] and Romakon PM 102 [15], were added as default membranes to the PyVaporation package in order to provide a researcher with a reference for the correct format of a *Membrane* directory.

### 7.4. Pervaporation Function

The information regarding the functions used to fit experimental diffusion curves (2D data) or diffusion curve sets (3D data) is stored in *PervaporationFunction* objects. To include the complex behaviour of permeance with the variation in the feed mixture composition and process temperature, Equation (12) is used to fit the experimental permeance data. The fits may be obtained for each component by using the *fit()* and *find_best_fit()* functions, which take a *Measurements* object as one of the arguments. *Measurements* objects, in turn, may be generated from *DiffusionCurve* or *DiffusionCurveSet* objects by calling corresponding *Measurements* methods. A detailed description of the fitting principles implemented in the package is given in the *Approximation of Permeance functions* section of the article. *PervaporationFunction*s may be saved as binaries and plotted, and are callable with composition and temperature as floats to calculate a corresponding permeance value. Obtained fits may be visualized with or without experimental data by using the *plot()* method. An example of creation and working with a *PervaporationFunction* object is presented in Figure 5.

### 7.5. Pervaporation

To model a diffusion curve or a process, a researcher needs to create a *Pervaporation* object from *Membrane* and *Mixture* objects. Modelling may be performed by calling a corresponding *Pervaporation* method. *Pervaporation* objects are capable of modelling either ideal or non-ideal processes.

There are two methods available for the estimation of a diffusion curve for a membrane at a given temperature based on *IdealExperiments*, *DiffusionCurveSet* objects or their combination. The *ideal_diffussion_curve()* method is designed to model a diffusion curve with the components’ permeance values independent of feed mixture composition on the basis of the information provided in the membrane’s *IdealExperiments* object. Given a composition range and process parameters, the system calculates selective-transport properties at each point in the assumption of permeance’s independency of feed mixture composition. The method returns a *DiffusionCurve* object that may be saved or used for further evaluation in the system. In terms of an ideal process assumption, the apparent energy of transport activation is also considered independent of the feed mixture composition. Example of the creation of an ideal diffusion curve with *Pervaporation.ideal_diffussion_curve()* is provided below (Figure 6):

The second method for modelling a diffusion curve at a specified temperature—*non_ideal_diffusion_curve()*—accounts for the permeance and apparent permeation activation energy dependencies on feed composition and takes a *DiffusionCurveSet* as an additional argument for modelling. If the *DiffusionCurveSet* object of a studied membrane contains diffusion curves at various temperatures, the method will use them as a basis for the evaluation of process dependency on temperature; however, if a single diffusion curve is specified, the method will assume the constancy of the apparent energy of the permeation value for a component over a considered composition range and will need an *IdealExperiments* object specified for the membrane for the calculation of its values. Example of the creation of a non-ideal diffusion curve is as follows (Figure 7):

By providing the initial permeances of the components at the beginning of the considered process, which are equivalent to the “dry” measured permeances of the components at the initial feed composition stated, a researcher may account for the influence of membrane swelling on the membrane’s performance. Equations (13) and (14) are used for the assessment of the membrane performance’s change with the change in the initial feed concentration of the process. Such an approach must be applied with care; however, it was proven to be able to describe real “wet” diffusion curves at various initial feed compositions based on the reference “wet” curve and the “dry” permeance values of the components at initial feed compositions of interest. An example of the code used for the modelling of Pervap 4100™ diffusion curves at 95 °C and different initial feed concentrations based on the curve fit illustrated in Figure 2a is presented below (Figure 8).

The modelling of the time-dependent pervaporation processes within the package may be performed by calling the *ideal_isothermal_process()*, *ideal_non_isothermal()*, *non_ideal_isothermal()* or *non_ideal_non_isothermal()* methods of *Pervaporation* class objects. Modelling is performed by consequent iterative calculations of the process parameters, while the process conditions at the next calculation step are evaluated based on the results obtained in the current step. The initial conditions necessary for the model’s obtainment, such as the initial feed amount, initial feed concentration, membrane area, permeate temperature or pressure, initial feed temperature or specified temperature program, are represented by *InitialConditions* objects. The temperature program may be specified under the initial conditions by adding a *TemperatureProgram* object, which supports polynomial, exponential and logarithmic types of temperature as a function of processing time. The parameters required to start the calculation are determined based on the initial conditions file, while the feed mixture amount, composition and temperature, defining the process mass balance, are determined based on the process mass and heat fluxes obtained at the current calculation step. The enthalpies of evaporation–condensation, mixture heat capacities and cooling heats are calculated according to Equations (15)–(20).

It should be noted that quasi-stationary processes within a pervaporation module operating at a constant feed mixture flux, where the process parameters depend on the dimension parameter of the module, may also be described by the implemented time-dependent processes by interpreting time as a dimensionless length and feed mass with the corresponding feed mass flux. It should be noted that this assumption remains valid until the feed mass flux values differ significantly from those of the retentate flux. The methods purposed for modelling time-dependent processes return an object of the *ProcessModel* class. The objects of this class contain information about the dependency of each considered process parameter on the processing time, initial conditions file and, when applied to non-ideal process modelling, which requires the fitting of experimental data with a *PervaporationFunction* object, permeance functions for each component used for modelling. The obtainment of a *ProcessModel* object for an example of an ideal isothermal process is presented below (Figure 9).

### 7.6. Process Model

The *ProcessModel* object contains information on the modelled time-dependent pervaporation process. Such objects store the feed temperature, feed and permeate compositions, feed mass, partial component fluxes and permeances, feed evaporation and permeate condensation heats throughout the processing time. These objects may be saved in a form of a directory with a specified structure, which contains a .csv file with all process parameters and binary *PervaporationFunction* and *InitialConditions* files. *ProcessModel* could be loaded from a directory with corresponding structure and contents to the system for further evaluation. The methods defined for the *ProcessModel* are similar to those defined for *DiffusionCurve* objects and allow a researcher to calculate basic derivative parameters, such as the separation factor, PSI or selectivity as functions of the processing time. Similar to the *DiffusionCurve* objects, the *plot()* method allows one to plot a graph of a specified process parameter as a function of the processing time.

### 7.7. Description of Partial Flux Calculation Algorithms

The calculation of the components’ partial fluxes under given process conditions is a vital part of modelling a pervaporation process. Partial fluxes in PyVaporation are calculated using Equation (2), while permeance values are fitted using Equation (12), or assumed to be constant. When the component’s partial pressure is assumed to be zero, the component permeance value under given conditions, along with the value of the saturated vapour pressure of the component over the separated mixture, is sufficient for partial flux value calculation. However, in the case where the influence of permeate temperature or pressure on the process is considered, the partial pressure of the component in the permeate is required for the establishment of the driving force value. As a component’s partial pressure is dependent on the permeate composition, which, on the other hand, is defined by flux values, the partial fluxes’ calculation is recursive. Depending on the type of the experimental setup, the partial pressure of the component may be defined with the permeate composition and either with an absolute pressure on the permeate side of the membrane or with the temperature of the permeate. In the case where constant pressure is maintained on the permeate side of the membrane using a vacuum pump, gas sweeping the permeate pressure may be picked as a defining parameter; however, when permeate is condensed in the sealed pre-vacuumed chamber, the permeate temperature may be used for the evaluation of a components partial pressure.

Partial flux calculation in the PyVaporation package is implemented in the *calculate_partial_fluxes()* method of the *Pervaporation* class. T and composition of the feed mixture, along with the required precision, must be specified for the calculation. The algorithm for the calculation is simple and is schematically represented in Figure 4. The precision parameter represents the acceptable difference between permeate compositions at consecutive calculation steps if a user specified the permeate pressure or temperature parameters. The calculation of the permeate pressure at a user-specified temperature is performed based on the NRTL model, so it is valid only at temperatures above the mixture’s freezing point. When the precision criterion is met, the system considers the calculation complete and returns partial flux values as a tuple. The default precision value specified for the calculation is 0.03%. A user may also specify the permeances of the components for the calculation; if they are not provided, the method calculates them based on the data provided in the *IdealExperiments* object of the *Membrane*. An illustration of the precision parameter’s influence on the calculated partial flux values is provided in Figure 10.

### 7.8. Approximation of Permeance Functions

One of the most challenging tasks in modelling a general pervaporation process is finding the type of function that could accurately describe the behaviour of the components’ permeance values with varying process conditions. Due to the absence of a thorough general theoretical description of the diffusion processes in polymers, and particularly in pervaporation, most of the suggested function types are based on the evaluation of experimental data. As mentioned above, there are a number of functions that have been successfully applied to describe permeance as a function of the feed composition [16,23,39], and temperature [1]; however, there are some cases [20,24] where the suggested approaches fail to describe the process with acceptable accuracy. For these reasons, a new general expression was suggested to fit permeance as a function of the two main process parameters—the feed mixture temperature and composition (Equation (12)). In order to include the membrane’s swelling history, the facilitation rate parameter [12] is introduced to adjust the form of the function based on the “dry” permeance value. The function presented in Equation (21) is used for fitting the component’s permeance value as a general function of the process temperature and feed composition in PyVaporation, and is represented in the *PervaporationFunction* class.
(21)$$P_{i}\left(x_{F},\;T\right)=\alpha \cdot FR\left(x_{F0}\right)\cdot exp\left(\sum_{i=1}^{n}\;a_{i}x^{i}-\frac{\sum_{i=0}^{m}\;b_{i}x^{i}}{T}\right)$$

The fitting with Equation (21) is a two-stage task; first of all, the order of inner-exponential polynomials must be determined (n and m) to finalize the function form, and then the coefficients of the function must be determined. The general fitting of the experimental data with Equation (21) in the package is realized by the *fit()* function, which requires user-specified n and m values as integers, and a set of experimental data points as a *Measurements* object, which is obtained from either *DiffusionCurve* or *DiffusionCurveSet* objects. The fitting is performed by *scipy.optimize.minimize()* [29] by means of the Powell minimization algorithm [43], with the objective function for minimization determined as a root-mean-square deviation. As a result, the *fit()* function returns a *PervaporationFunction* object, defined with user-specified n and m values, and the coefficients obtained during optimization. The suggestion of n and m values is implemented in the scope of the *find_best_fit()* function of the package. The function requires only a *Measurements* object and suggests maximal n and m values based on the size of the fitted dataset; after the suggestion, the function fits the data by means of the *fit()* function with n and m in the range from 0 to the suggested maximal polynomial order values and, as a result, returns the *PervaporationFunction* with the lowest objective function value in relation to the fitted dataset. A user may override suggestions by specifying the n and m values directly in *find_best_fit()* arguments. It should be noted that the n and m values may significantly affect the extrapolation of the permeance values; thus, the fits should be verified by the researcher prior to usage.

An example of fitting experimental data, illustrated in Figure 1a, using the *find_best_fit()* function is provided below (Figure 11). The facilitation rate parameter is calculated only when modelling a non-ideal process or diffusion curve based on the provided permeances of the component measured at the considered initial feed composition and the permeances of the base diffusion curve calculated using the corresponding *PervaporationFunction* (Equation (14)).

### 7.9. Evaluation against Experimental Data

Several examples of applying PyVaporation to the fitting of experimental data and modelling diffusion curves and processes provided above illustrate the suitability of the instrument for a number of the most commonly solved tasks. In order to illustrate further possibilities of the developed framework and the theoretical approach behind it, the modelling of a non-ideal isothermal process and evaluation of the model against real experimental data were performed. Due to the abundance of data on hydrophilic pervaporation, the model was evaluated only for application to this pervaporation type. However, the general theoretical approaches implemented in the package may also be applicable for the description of hydrophobic pervaporation.

To illustrate the predictive power of PyVaporation, the experimental data provided by Yave [24] on the performance of Pervap 4101 in relation to water/ethanol mixture separation was used to model the isothermal process performed by Thiess et al. [44]. Using an independent information source as a basis for modelling is one of the key advantages of using a suitable modelling approach. Thus, by comparing the data obtained based on the membrane performance reported by Yave and experimental data obtained by Thiess et al., the applicability of PyVaporation as a tool for solving a typical research task may be evaluated. The code used to obtain the model is provided in Figure 12. An illustration of the modelling results along with the corresponding RMSD values is provided in Figure 5.

In both articles, the performance of Pervap 4101 membranes was studied thoroughly at 95 °C in relation to the separation of water/ethanol mixtures. It should be noted that the data provided in [24] were given for an initial feed comprised of 16 wt% water, while the experimental process in [44] was conducted starting with 10 wt% water in the feed. Due to the fact that the performance of Pervap 4101 is dependent on the initial feed composition, the introduction of the FR parameter is necessary to describe the modelled process correctly. For these reasons, initial permeance values were stated to obtain the appropriate value of the FR parameter.

A comparison of the modelling results with experimental data showed a high level of accuracy, illustrated by low RMSD values. Lower RMSD values for the water fraction (Figure 5a) in the feed may be associated with higher measurement accuracy when compared with flux values (Figure 5b). The evaluation of the model obtained using the PyVaporation package against the model proposed by Thiess et al. [44] allows the conclusion that both models provide results of similar accuracy, while PyVaporation utilized data from an independent source [24], with a single experimental point being required for modelling for the calculation of the FR parameter inside the *non_ideal_isothermal_process() method.*

For further evaluation of the PyVaporation package’s performance, an isothermal experiment at 68.8 °C involving the separation of the water/ethanol mixtures with an initial ethanol concentration of 8.7 wt% was performed using Romakon-PM 102 membranes. The effective membrane area in the cell used was 25.5 cm^2^, the initial feed amount was 60 g and the absolute permeate pressure was measured to be 2 ± 1 kPa. The data on the performance of Romakon-PM 102 reported in [15] were used for modelling, and a description of the experimental setup and methods used is provided in the Appendix A. The code used to model the experiment is given in Figure 13, and the evaluation of the model obtained with PyVaporation against the experimental data is illustrated in Figure 6.

Both methods implemented in the package for ideal and non-ideal modelling provide similar results due to the constancy of the components’ permeance values in the considered composition range. The relatively low RMSD value illustrated in Figure 6 indicates PyVaporation’s accuracy and predictive ability.

Additionally, the modelling of a non-ideal non-isothermal process with PyVaporation was evaluated against the experimental data reported by Chang et al. [45] on the pervaporation separation of water/ethanol mixture. Time-dependent process modelling implemented in PyVaporation was treated as quasi-stationary by interpreting the feed amount as a mass flux and time as the corresponding linear dimension. The initial feed mass flux was calculated from the volume flux values reported in the work by multiplying it by the corresponding mixture density value. The experimental data on the membrane performance reported by Chang and colleagues were used as the basis for modelling. The feed temperature reached during interstage heating was averaged throughout the obtained experimental values and assumed to be 95.3 °C. The code used to perform modelling is provided in Figure 14, while the modelling results are illustrated in Figure 7, along with the corresponding experimental values.

The comparison between the modelled and experimental temperatures within the membrane node illustrated in Figure 7a allows us to conclude that the absolute difference between them does not exceed 5.5 °C. The predicted concentration of the feed mixture as a function of membrane area, illustrated in Figure 7b, indicates a good correlation with experimental data, which is characterized by an RMSD value of 4.39%. Judging by this, the performance of PyVaporation is not inferior to that of the original model proposed by the authors in [45] for application to modelling adiabatic non-ideal pervaporation process.

## 8. Conclusions

An exponential–polynomial form of the permeance equation is suggested for the description of highly non-ideal pervaporation separation and was proven to show acceptable accuracy in the case of hydrophilic pervaporation. The polynomial characteristic of the inner exponential term allows accounting for local minima and maxima of permeance as a function of feed composition, which may occur due to the overlapping and/or transition between the transport mechanisms throughout the considered composition range. The polynomial representation of the apparent transport activation energy term allows accounting for its dependency on feed composition, as observed in some cases. The introduction of the facilitation rate parameter as an instrument to describe the influence of the initial feed composition on the form of the membrane’s diffusion curve is demonstrated to be a valid assumption with relatively strong predictive ability. The methodology of “dry” and “wet” diffusion curve measurement is discussed and suggested as a way to calculate the facilitation rate parameters. Fitting algorithms implemented as part of the PyVaporation package allow a researcher to fit experimental data with the suggested permeance equation and use it for modelling hydrophilic pervaporation processes. The package was demonstrated to model real experimental data, based on independent sources of membrane performance data, with high accuracy (RMSD: 5–9% flux, 1–5% separated mixture concentration). A format for storing and sharing the information on pervaporation membranes and modelling results is proposed for the unification of the reported data.

A comprehensive, freely available Python-based functional instrument for modelling and studying pervaporation processes was developed and introduced. The performance of the developed package was evaluated against a number of experimental cases of hydrophilic pervaporation experiments, which proved its applicability. The comparison of PyVaporation’s modelling ability with several modelling approaches reported in the literature indicates that, in most cases, the package may be used to obtain similar modelling results with high predictive accuracy, and could have been successfully used instead of them.

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
