# Peer review of "PyVaporation: A Python Package for Studying and Modelling Pervaporation Processes"

_membranes, 2022, doi:10.3390/membranes12080784_

Round 1

Reviewer 1 Report

The main contribution of this work is to evaluate the theoretical background of the solution, its

 applicability and limitations of freely available Python package for pervaporation processes. I found the paper very interesting and innovative. It is very well articulated, data are well described and reported. I suggested that this manuscript could be published in Membranes after major revision.

1. Authors did not provide important quantitative analysis in the Abstract and Conclusion sections. Therefore, the abstract is insufficient to support the proposed methods and their background. In order to clearly emphasize the research work, the Abstract and Conclusion sections need to be rewritten. Authors are highly encouraged to include the point-by-point highlights of this article.

2. Graphical abstract should be included in the manuscript.

3. Did you adjusted any parameters of this freely available Python package for pervaporation processes? Or you only explore the applicability of this Python package for pervaporation processes? Authors should clarify this point.

4. From line 21-118, a lot of meaninglessness descriptions in this part as you only want to explain the background of difficulties in the existing pervaporation processes. Please briefly state the background.

5. Can you share your source code to other researchers?

Author Response

Dear Reviewer,

Thank you very much for your suggestions and recommendations.

  1. We have added an overview of general quantitative characteristics of the models obtained with the package in the Introduction and Conclusion sections. please see the revised manuscript.
  2. We initially added the graphical abstract for the manuscript, we have also added the image to the revised version.
  3. No parameters were adjusted, the package was used as released. The package takes the data on membrane performance as an input and has a number of functions which could be used to obtain various models. Most of the code examples may be executed using supplementary code-examples.ipynb in Jupiter notebook, which is also available in the main branch of the package's repository.
  4. Thank you for your suggestion. We think that it is extremely important to discuss the peculiarities of the process and scientific background in detail to provide readers with an understanding of the solution's applicability. 
  5. The link to the source code of the solution is provided in the reference [31] (https://github.com/Membrizard/PyVaporation) of the article. The source code used for the preparation of the release described in the article is also provided in the supplementary data. 

Reviewer 2 Report

Dear authors,

I have reviewed the paper membranes-1864039, entitled PyVaporation: a Python package for studying and modeling pervaporation processes. In this article, the authors present a Phyton package to perform the modeling and simulation of pervaporation processes based on physical-chemical and membrane technology principles. The modeled data is compared to experimental data obtained from the literature.

The article is very interesting and addresses an emerging and current area. However, there is a specific issue that I would like to point out.

In model validation section, all comparisons between the developed algorithm and experimental data were of hydrophilic pervaporation processes. Considering that pervaporation processes and membranes can be divided into three main groups (hydrophilic, organophilic or hydrophobic, and target-organophilic pervaporation), presenting validation examples of only hydrophilic pervaporation suggests that the other pervaporation types may not be modeled or simulated with a similar precision/performance.

I would also like to ask the authors: have hydrophobic and target-organophilic pervaporation been modeled using the PyVaporation package? If yes, please, provide the validation in the same way as done with hydrophilic pervaporation experiments. If not, I advise stating clearly throughout the paper that the package and algorithm are only validated for hydrophilic pervaporation.

Considering the above, I believe it is essential for the paper that the authors provide validation examples of the three pervaporation types and also address the specificities of each type and the possible impacts on the developed model.

In addition, the English of the paper should be revised to allow for better reading. There are sections that are difficult to understand due to the language.

Please, refer to the attached pdf file for specific recommendations.

Author Response

Dear Reviewer,

You are indeed right, we only checked the application of PyVaporation to the modelling of hyprodphilic pervaporation processes. It is undoubtedly important to state that the package was validated against only hydrophilic pervaporation experiments; we have done so in the Evaluation against experimental data and Abstract sections.

We would like to thank you for your questions and comments throughout the manuscript, please see our answers to them (the references are to the lines of the original submission):

lines 16-17.

“The authors could provide some comparison regarding pervaporation types, e.g., average accurracy and stantdard error of the model regarding experimental data for hydrophilic pervaporation, hydrophobic pervaporation, and target-organophilic pervaporation, among other parameters.”

Thank you for the suggestion, we have added the quantitative characteristics of the models obtained for hydrophilic experiments into the Abstract and Conclusions section, please see the revised manuscript.

line 18.

“Please, avoid repeating terms already present in the title in the keywords.”
We have revised the keywords accordingly, please see the revised manuscript

lines 53-54.

“Chemical affinity, linked to sorptive properties of the material and the feed component”

We agree to that, however we believe our phrasing is sufficient.

line 77 - changed to “membrane” as suggested.

line 125 - changed to “carried out” as suggested.

line 133 - changed to “This work is aimed to” as suggested.

line 157.

“This assumption is correct in pervaporation applying vacuum in the permeate stream. What about other pervaporation processes? A model using a chemical potential difference would be more general and comprehensive.”

We, indeed, agree with this statement, however, in general, even when thermal or gas sweeping pervaporation is considered, partial pressures of the components on the permeate side play a deterministic role in the driving force value. As most of the industrial and laboratory process are implemented in vacuum or gas sweeping types, we find this assumption general enough to be useful to a wide variety of applications and at the same time this assumption provides enough simplicity for the calculations. 

lines 160-161.

“Selectivity is regarded as an intrinsic membrane property relative to the feed components (independent of the their composition), while the separation factor depends on the specific concentrations of the feed components.

What are the aspects relevant to the model of using one or the other or both parameters?”

The ideal selectivity, which is defined as the ratio of components’ permeances or permeability coefficients measured in terms of individual components is undoubtedly is not dependent on the feed mixture composition. There is , on the other hand, the real selectivity terms, which is defined by the same equation, however measured during the experiment of a considered mixture separation, and its value is often strongly dependent on the feed composition. Both numerical characteristics separation factor and real selectivity are used for the quantitative representation of the membrane’s separation performance. While separation factor is similar to “relative volatility ” it is useful for comparison of the pervaporation process to analogous distillation ones, and is widely used for characterisation of the membranes’ separation performance, the real selectivity values also illustrative and may be more easily used for the calculation of the fluxes of the components fluxes if necessary,

line 169.

“Present and explain this term. It was not addressed.”

It is the separation factor value defined with equation (4)

lines 201-205.

“Although considering that the pressure in permeate side are usually very low, can Dalton's law be considered as suitable to determine the partial pressures of the components of permeate stream and, consequently, their molar fractions in the permeate? Have the authors assessed this possibility, or even using real gas approaches?”

Thanks for the question. Usually in terms of low values of pressure the non-ideality of permeate vapours could be neglected without drastic effect on the model. However, when relatively high pressures at the permeate side are considered a user must handle this assumption with care. That is exactly why we are explaining this assumption in detail, to state the applicability of the developed solution. We considered including the description of permeate vapours non-ideality, however, this did not provide significant increase in the models’ accuracy, while making the usage of the package difficult. If necessary a user may alter the code to account for permeate vapours non-ideality if needed. Even without the usage of real gas approaches the models obtained show an extremely good correlation with the experimental data

line 236.

“What is the approximate cutoff value or the pressure transition range between these two situations? Or what is considered in terms of individual substances?”

The typical values of the permeate absolute pressure in terms of pervaporation are 1-3 kPa, sometimes up to 4 kPa are observed. The transition between these two situation is dependent on the type of the mixture (degree of non-ideality of vapours) and on the driving force of the process, thus there is usually no universal value for the transition.
Usually, the permeate pressure may be considered low, when its significantly lower  (not more than 5-10 %) than the driving force value.

lines 270-272.

“Here, the authors addressed hydrophilic pervaporation. What about organophilic and target-organophilic pervaporation?”

We have indicated that the package was evaluated only in application to modelling of hydrophilic pervaporation, please see the revised manuscript.

line 425.

“Question: only binary mixture can be modeled using PyVaporation? Or is this just an example?”

Thanks for the question! For now only binary mixtures are supported, due to our inability to thoroughly evaluate and the performance and prove the applicability of the solution to multicomponent mixtures separation, as there is a very small amount of acceptable experimental data points reported in the literature. We are working on the broadening the package functionality, however release 1.1.4 does support only binary mixtures.

line 426.

“Why was the NRTL model chosen? Can the user choose another thermodynamic model?”

The NRTL model was chosen due to relative ease of implementation, fast calculation of the parameters and acceptable accuracy while describing the built-in mixtures.
A user may use other type of modelling, but for that it is necessary to alter the code of the solution. The solution was written in the way that only one function is responsible for calculation of the saturated vapour pressures - get_nrtl_partial_pressures() in the Mixture module of the solution. If a user wants to alter the model used for these calculation he/she should implement the desired model in the function. We are also working on the broadening of the implemented thermodynamic models range, including on the ability to use third-party packages, however this is not part of the described release (1.1.4)

lines 473-474.

“What is the impact of using this assumption when  compared to a real process?

Perhaps could be possible for the user to set a permeate pressure, since this will also affect the process?”

The calculation of permeances in the assumption of zero permeate pressure inside the Diffusion curve class instances is performed only if neither permeate pressure or temperature is stated. If they are stated in the input they are accounted for.

lines 613-614.

“Is it feasible to implement these tools in the package to do the calculations and store the calculated values?

This would be an excellent tool to help analyze process performance.”

Thank you very much for the comment. We aimed to develop a general solution that could help other researchers relatively easy describe, model and analyze their pervaporation data as we suffered with that a lot as well. Actually, these are relatively easy arithmetic calculations and they are performed only if corresponding methods are called based on the data stored in the object.

For example:
example_diffusion_curve.get_psi() will calculate PSI at every concentration point specified in the DiffusionCurve object and will return them in a form of a list.

So this does not drastically affect productivity.

line 659.

“All examples shown in this section correspond to hydrophilic pervaporation processes. I strongly advise the authors to also present modeling and comparison of hydrophobic and/or target-organophilic pervaporation data, to show the comprehensiveness of the model. In the current presentation, only conclusions about hydrophilic pervaporation can be drawn.”

We strongly agree, the Abstract and Conclusion sections were revised, indicating that evaluation was performed only in the case of hydrophilic pervaporation.

line 788.

“This section should be shorter and go directly into the main conclusions of the study, rather than repeating some results”

We have revised and shortened the section, please see the revised manuscript.

lines 790-791, 803, 812-813.

“Only for hydrophilic pervaporation systems.”

Agreed, added the note to the sentences, please see the revised manuscript

Round 2

Reviewer 1 Report

Authors addressed all my concerns, now I think this paper can be accepted by Membranes.

Author Response

Thank you very much for your suggestions.

Reviewer 2 Report

Dear authors,

I have reviewed the revised version of the paper and I believe it has improved a lot with the changes made. As final suggestions, I would recommend:

- use the term 'hydrophilic pervaporation' in the title, or add it to the keywords;

- The explanation "The ideal selectivity, which is defined as the ratio of components’ permeances or permeability coefficients measured in terms of individual components is undoubtedly is not dependent on the feed mixture composition. There is , on the other hand, the real selectivity terms, which is defined by the same equation, however measured during the experiment of a considered mixture separation, and its value is often strongly dependent on the feed composition. Both numerical characteristics separation factor and real selectivity are used for the quantitative representation of the membrane’s separation performance. While separation factor is similar to “relative volatility ” it is useful for comparison of the pervaporation process to analogous distillation ones, and is widely used for characterisation of the membranes’ separation performance, the real selectivity values also illustrative and may be more easily used for the calculation of the fluxes of the components fluxes if necessary." could be adapted and added to the paper to render the work more comprehensive.

- The authors explained: "Thanks for the question. Usually in terms of low values of pressure the non-ideality of permeate vapours could be neglected without drastic effect on the model. However, when relatively high pressures at the permeate side are considered a user must handle this assumption with care. That is exactly why we are explaining this assumption in detail, to state the applicability of the developed solution. We considered including the description of permeate vapours non-ideality, however, this did not provide significant increase in the models’ accuracy, while making the usage of the package difficult. If necessary a user may alter the code to account for permeate vapours non-ideality if needed. Even without the usage of real gas approaches the models obtained show an extremely good correlation with the experimental data." I suggest adding this information to inform the readers that both ideal and real gas approaches can be considered both suitable and the use of one or antoher should be subject to the researcher's own judgement.

- The authors explained: "The typical values of the permeate absolute pressure in terms of pervaporation are 1-3 kPa, sometimes up to 4 kPa are observed. The transition between these two situation is dependent on the type of the mixture (degree of non-ideality of vapours) and on the driving force of the process, thus there is usually no universal value for the transition. Usually, the permeate pressure may be considered low, when its significantly lower (not more than 5-10 %) than the driving force value." This is an interesting information and it could be added to the paper to give reference values to researchers in the area.

- And finally, I recommend the possibility of using different thermodynamic models to calculate the data in future releases. I understand that some thermodynamical models are very complex and difficult to implement, but, if possible, this would render this package extremely powerful.

I would also like to congratulate the authors for the development of this package. It is a very innovative and interesting tool to help researchers in the field of pervaporation, a very challenging area.  I sincerely hope to see this package expanded to cover also hydrophobic and target-organophilic pervaporation processes in the future.

Author Response

Dear reviewer,

Thank you for such a quick reply.

  • We would like to leave the title and keywords as is. Despite the fact that we did not validate the model against hydrophobic and target organophilic pervaporation experiments, the similarity of approaches for modelling these processes suggests that the model will be applicable for the description of such processes [https://doi.org/10.1081/spm-100102985]. We would like to encourage readers to try and use the instrument for the description of their experimental data of different types.
  • Thanks for your suggestion, we have added an adapted version of an explanation to the text, just before equations 3 and 4. Please see the revised manuscript.
  • Thank you for your suggestion we have adapted the response and added it to the corresponding section.
  • Thank you for your suggestion we have adapted the response and added it to the corresponding section.

Thank you for the suggestion, we encourage you to visit the GitHub page of the project to look out for updates. (https://github.com/Membrizard/PyVaporation).
We are currently working on the integration of UNIFAC/UNIQUAC models, which are not that hard but challenging to implement.